# Accelerated Deep Active Learning with Graph-based Sub-Sampling

## Abstract

Past years have witnessed the fast and thorough development of active learning, a human-in-the-loop semi-supervised learning that helps reduce the burden of expensive data annotation. Diverse techniques have been proposed to improve the efficiency of label acquisition. However, the existing techniques are mostly intractable at scale on massive unlabeled instances. In particular, the query time and model retraining time of large scale image-data models is usually linear or even quadratic in the size of the unlabeled pool set and its dimension. The main reason for this intractability is the iterative need to scan the pool set at least once in order to select the best samples for label annotation.

To alleviate this computational burden we propose efficient Diffusion Graph Active Learning (DGAL). DGAL is used on a pre-computed Variational-Auto-Encoders (VAE) latent space to restrict the pool set to a much smaller candidates set. The sub-sample is then used in deep architectures, to reduce the query time, via an additional standard active learning baseline criterion. DGAL demonstrates a query time versus accuracy trade-off that is two or more orders of magnitude acceleration over state-of-the-art methods. Moreover, we demonstrate the important exploration-exploitation trade-off in DGAL that allows the restricted set to capture the most impactful samples for active learning at each iteration.

## 1 Introduction

Deep learning has provided unprecedented performance in various semi-supervised learning tasks ranging from speech recognition to computer vision and natural language processing. Deep Convolutional Neural Networks (CNN), in particular, have demonstrated object recognition that exceeds human's performance. However, this success comes with the requirement for massive amounts of labeled data. While data collection at large scale has become easier, its annotation with labels has become a major bottleneck for execution in many real-life use cases Settles (2009); Ren et al. (2021). Active learning provides a plethora of techniques that allow to select a set of data points for labeling which optimally minimize the error probability under a fixed budget of a labeling effort (see Settles (2009) for review). As such it is a key technology for reducing the data annotation effort in training and developing semi-supervised models.

One of the key caveats in active learning, preventing it from being used on an industrial web-scale, is the computational burden of selecting the best samples for annotation at each step of active learning. This complexity is rooted in a variety of important criteria that need to be optimized in active learning. Referred to as the 'two faces of active learning' Dasgupta (2011), the most common selection mechanisms can be categorized into two parts: uncertainty (e.g. Lewis & Gale (1994)) and diversity sampling (e.g Ash et al. (2019a); Sener & Savarese (2017a)). The intuition of the former is to select query points that improve the model as rapidly as possible. The latter exploits heterogeneity in the feature space, sometimes characterized by natural clusters, to avoid redundancy in sampling. The combination of the uncertainty and diversity criteria has been an important subject of recent works (Yuan et al., 2020; Zhu et al., 2008; Shen et al., 2004; Ducoffe & Precioso, 2018; Margatina et al., 2021; Sinha et al., 2019a; Gissin & Shalev-Shwartz, 2019; Parvaneh et al., 2022; Huijser & van Gemert, 2017; Zhang et al., 2020).

Optimizing for uncertainty or diversity (or both) may require scanning all data at least once, and sometimes even applying methodology with computational costs that scales quadratic or more in the unlabeled pool size (e.g. Bodó et al. (2011); Tong & Koller (2001); Sener & Savarese (2017a)). An active learning cycle repeats the time-consuming model (re-)training and query selection process multiple times. In many cases this cumbersome repetitions render active learning as impractical on real large scale data sets. In fact, even a single feed-forward process in a deep learning network for uncertainty calculation (e.g. Gal et al. (2017)) may impose a significant delay in query time. In fact, it may scale non-linearly in the dimension of the data (e.g. number of pixels) for each query candidate in many deep network architectures, where fully connected layers exists. To this end, only a few approaches have been suggested to overcome this bottleneck in the context of deep learning. Most notable is SEALS (Coleman et al., 2022a), which improves the computational efficiency of active learning and rare class search methods by restricting the candidate pool for labeling to the nearest neighbors of the currently labeled set instead of scanning over all of the unlabeled data. The restricted set is given as an input to the task classifier for a second selection step, and using a second basic uncertainty criterion a final query set is selected for annotation. We identified that for that SEALS criterion for selecting $k$-nearest neighbors to the restricted pool does not address the diversification criterion in query selection. In fact, it also does not capture the exploration-refinement transition which improves active learning tremendously. We therefore propose a different algorithm for selection of the restricted pool set based on a graph diffusion algorithm inspired by Kushnir (2014); Kushnir & Venturi (2023). We refer to it as Diffusion Graph Active Learning (DGAL). In DGAL, the proximity graph is computed **only once** for a latent Variational Auto Encoder (VAE) Kingma & Welling (2013) representation space which is trained once prior to the annotation cycles. This graph representation is then used for a label diffusion process to select the most diversified and uncertain candidates in time that is linear in the data size. The graph, computed only once, allows faster linear query time in each diffusion process, unlike SEALS (Coleman et al., 2022a), whose query time keeps growing by a factor dependant linearly on the number of neighbors $k$.

As seen in the overview of our method in figure 1, our method is comprised of two components: a representation component in which a VAE training is occurring once and used to generate a graph representation that is then shared in the iterative stage of the active learning component. The same graph is used throughout the active learning iterations, in the second component, to select a restricted and small set of candidates which is fed into the classification neural net for the final query selection (e.g. uncertainty sampling (Lewis & Gale, 1994), margin (Scheffer et al., 2001), etc.). As shown in our paper the restriction t a smaller set via our graph construction accelrate active learning which achieving state of the art or better accuracy.

The graph-based diffusion algorithm used in DGAL enhances important criterion in active learning referred to as the exploration-exploitation (or exploration-refinement criterion). Exploration addresses a stage in active learning in which data is sampled and annotated to first map decision boundaries in the data. On the other hand, exploitation takes the so far detected boundaries and samples points around it to better localize it. At early stages of AL an exploratory strategy typically yields better gains in accuracy over boundary refinement ones. However, once all boundaries are detected, typically a refinement stage provides better accuracy gains over further exploration. This important trade-off is not leveraged in standard simple AL criteria such as probabilistic uncertainty sampling or diversification per-se. In particular, a k-nearest neighbors to the training set (Coleman et al. (2022a)) also does not reflect this trade-off. We demonstrate in this paper that our combined graph methodology for pool set restriction improves AL query time, and scales it to large scale data sets. We summarize our contributions below:

- We propose DGAL, a two-step active learning algorithm, that starts with restricting the pool set to a smaller set using a graph-diffusion process, and then uses the restricted set to perform deep active learning efficiently.

- DGAL achieved orders of magnitude acceleration in query time compared to state-of-the-art (SOTA) active learning schemes, while maintaining most competitive accuracy.

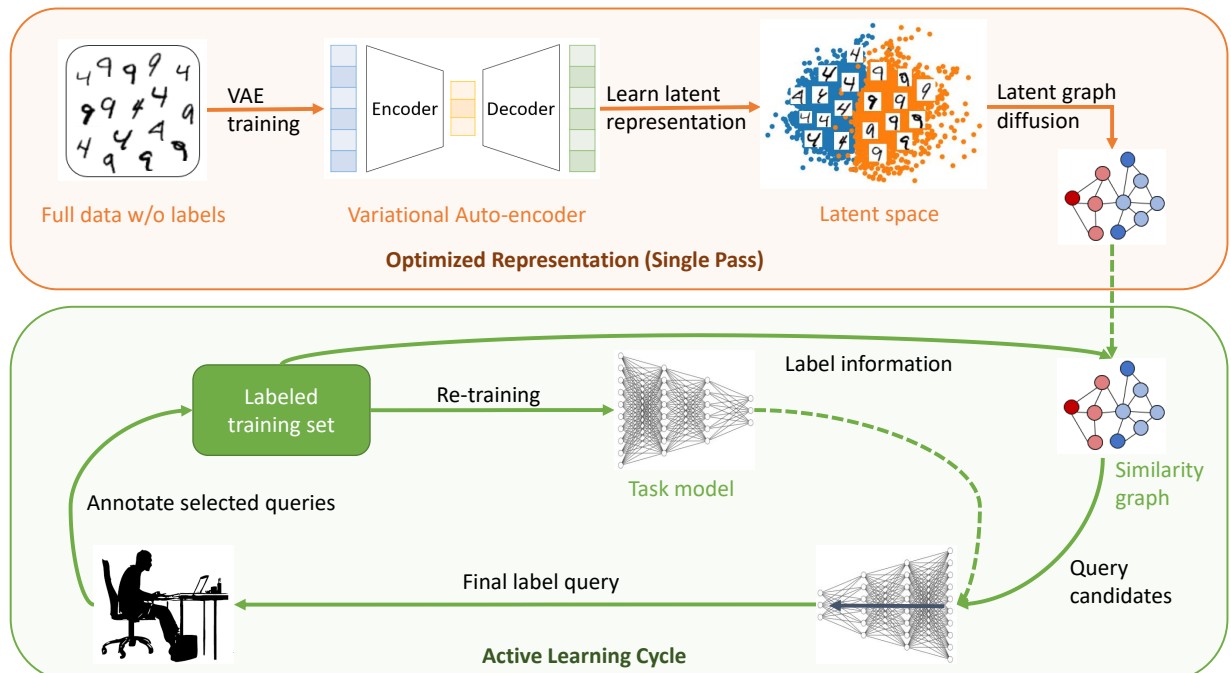

Figure 1: **Overview of our DGAL approach**. A Representation component that involving a single pass on the data for deriving a VAE-based latent graph representation. An Active Learning iterative component which uses the VAEs latent representation graph in conjunction with existing label information to select a restricted set of query candidates. The set of candidates is fed into a task neural net, where a second criterion is used to select the final label queries for annotation.

## 2  Related research

**Efficient Active Learning.** The application of active selection criteria on large pool sets and its computational costs has been an impediment for the application of active learning of industrial large scale data. This realization has motivated the development of more compact, efficient AL algorithms to cope with large-scale data sets/models .

In SEALS (Coleman et al., 2022a) the candidate pool is restricted to the k-Nearest Neighbours (KNN) of so-far-labeled instances, to avoid the computational burden of batch selection with large pool sets. However, the KNN set is typically very similar to the already labeled training data and therefore its information content for improving the classifier is low. This lack of diversification and exploration in SEALS yields sub-optimal accuracy. Moreover, the number of nearest neighbours being fed later to the tasks neural net is growing by a factor of $k$ on each query step. This leads to an increasingly higher query time when the restricted pool is fed into the task network. Xie et al. (2021) propose knowledge clusters extracted from intermediate features over pre-trained models by a single pass. Ertekin et al. (2007); Doucette & Heywood (2008) proposed to actively subsample unlabeled candidates under imbalanced datasets in general learning schemes, different than deep learning. Generating informative samples (Mayer & Timofte, 2020; Huijser & van Gemert, 2017; Zhu & Bento, 2017) saves time from unlabeled data acquisition aspect. Small but well-trained auxiliary models have been used to select data in order to reduce the computational cost of extracting feature representation from the unlabeled pool (Yoo & Kweon, 2019; Coleman et al., 2019).

**Graph-based Semi-Supervised Learning (GSSL)** Semi-supervised learning (SSL) (Zhu, 2005) exploits the information of both the labeled and unlabeled datasets to learn a good classifier. As a form of SSL, active learning automates the process of data acquisition from a large pool of unlabeled dataset for annotation to achieve the maximal performance of the classifier (usually) under a limited budget. GSSL (Zhu, 2005; Song

et al., 2022; Zha et al., 2009) is a classic branch of the SSL that aims to represent the data in a graph such that the label information of the unlabeled set can be inferred using the labeled data. A classic and solid technique is the label propagation or Laplace learning (Zhu et al., 2003), which diffuses label information from labeled set to unlabeled instances.Notably, the computation complexity of typical label propagation algorithms is only linear in the size of the data, which renders them as an efficient choice for learning.

The success of label propagation hinges on an informative graph that retains similarity of the data points. Due to the volatile property of image pixels, i.e., unstable to noise, rotation, etc., feature transformations Lowe (1999); Bruna & Mallat (2013); Simonyan & Zisserman (2014) are usually applied to build good quality graphs. Past research Doersch (2016); Kingma et al. (2019); Mei et al. (2019); Miller et al. (2022); Calder et al. (2020) has shown that the VAE can generate high-quality latent representation of data for feature extraction and similarity graph construction. We utilize these properties in our DGAL framework.

**Generative Models in Active Learning.** Deep generative models have been used to learn the latent representation of data in both semi-supervised and unsupervised learning (Kingma et al., 2019; 2014). Except for constructing similarity graph in GSSL, they can also be exploited to generate adversarial data/models for more efficient and robust AL. For example, Sinha et al. (2019a) proposed a task-agnostic model that trains an adversarial network to select unlabeled instances that are distinct from the labeled set in the latent space of a Variational Auto-Encoder (VAE). The DAL (Gissin & Shalev-Shwartz, 2019) selects samples in a way to make the labeled and unlabeled set hard to be distinguished in a learned representation of the data. Miller et al. (2022) embeds the Synthetic Aperture Radar (SAR) data in a latent space of VAE and apply a GSSL method on constructed similarity graph in this feature space. Pourkamali-Anaraki & Wakin (2019) aims at finding a diverse coreset in a representative latent space using K-Means clustering.

## 3 Problem setup and preliminaries

### 3.1 Active Learning

We consider a data set $D \in R^d$ of cardinally $[n]$. $D$ can split into a labelled set $D_l$ which represents a labeled subset of $D$, and an unlabeled subset $D_u$. $C$ denotes the number of classes. Fixing a batch size of $B$, we seek for the most 'informative' subset $D^\star$ to be annotated from an unlabeled pool set given a limited budget for annotation.

**Active Learning Problem Statement**. Let $f_\theta$ be a classifier: $f_\theta : \mathbb{R}^d \rightarrow \Delta_C$, ($\Delta_C$ denotes the space of probability measures over $[C] = \{1, ..., C\}$).Assume

---

**Algorithm 1** A general active learning algorithm

**Input:** Labeled data $D_l$, unlabeled pool $D_u$, budget size $B$, maximum round $R$, task model $f_\theta$
Initialize $D_l$ by random sampling and train $f_\theta$ on
**for** $i = 1$ **to** $R$ **do**
    query $D' \leftarrow \boldsymbol{QueryStrategy}(D_l, D_u, B, f_\theta)$
    **update** $D_l \leftarrow D_l \bigcup D'$, $D_u \leftarrow D_u \setminus D'$
    **train** model $f_\theta$ on new labeled pool $D_l$
**end for**

---

data are sampled *i.i.d.* over a space $\mathcal{D} \times [C]$, denoted $\{\mathbf{x}_i, \mathbf{y}_i\}_{i \in [n]}$.We would like to find a subset $D^\star \subset \mathcal{D}$ of cardinality $B$ such that

$$D^\star = \underset{D':|D'|<B}{\arg\min} \; \mathbb{E}_{\mathbf{x},\mathbf{y} \in D'}[\mathbb{I}\{\underset{c \in [C]}{\arg\max} f_c(\mathbf{x}) \neq \mathbf{y}\}] \tag{1}$$

gives the minimum expected error. We consider the multi-step active learning in algorithm 1.

### 3.2 VAE-based data representation

Latent variable models such as the VAEs are well-acknowledged for learning representations of data, especially images. VAE is a generative neural network consisting of a probabilistic encoder $q(\mathbf{z}|\mathbf{x})$ and a generative model $p(\mathbf{x}|\mathbf{z})$. The generator models a distribution over the input data $\mathbf{x}$, conditioned on a latent variable $\mathbf{z}$ with prior distribution $p_\theta(\mathbf{z})$. The encoder approximates the posterior distribution $p(\mathbf{z}|\mathbf{x})$ of the latent variables $\mathbf{z}$ given input data $\mathbf{x}$ and is trained along with the generative model by maximizing the evidence lower bound (ELBO):

$$ELBO(\mathbf{x}) = E_{q(\mathbf{z}|\mathbf{x})}[\log(p(\mathbf{x}|\mathbf{z})) - KL(q(\mathbf{z}|\mathbf{x})||p(\mathbf{z}))$$

where KL is the Kullback–Libeler divergence and $\log(p(\mathbf{x})) \geq ELBO(\mathbf{x})$.

To get a representative latent space from data (without label information), we use a ResNet18 (He et al., 2016) based encoder for large datasets and a CNN-based VAE for all other datasets. In the first term, $E_{q(\mathbf{z}|\mathbf{x})}[\log(p(\mathbf{x}|\mathbf{z}) = \frac{1}{n}\sum_i^n (\mathbf{x}_i - \mathbf{x}_i')^2$ where $\mathbf{x}_i'$ is the $i$-th (out of $n$) reconstructed image. We use the MSE loss for the first term, i.e. the reconstruction loss. Next, we construct a proximity graph from the latent representation to be used within a graph diffusion process and label acquisition.

Figure 2 visualizes a 2D projection of a CNN-VAE 5D representation of digits '4' and '9' from the MNIST dataset. To demonstrate the important semantics of this representation for active learning, we focus on the decision boundary between digits. we observe samples of '4's that are similar to samples from '9' class along the boundary.

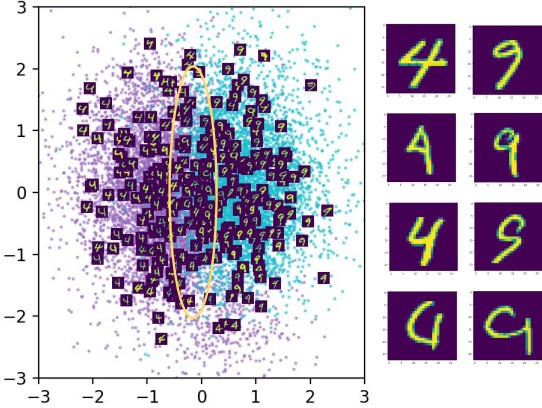

Figure 2: A VAE representation of digits '4' and '9' from MNIST.

## 4 Methodology

Below we explain the various components of VAE-DGAL, and then connect them to provide the overall DGAL scheme. Active learning is used in DGAL in a similar scheme to what is proposed in algorithm 1. However, we are using an active criterion in two components: first, we use a very efficient graph-diffusion-based selection criterion Kushnir & Venturi (2023) in the VAE latent space to restrict the pool set to a much smaller set of candidates. In the second component we use a second acquisition criterion to select from the set of query candidates the final query set for annotation. The second selection criterion bay be computationally intensive due to its nature, but also because it requires a feed-forward transmission. However, when applied to a smaller, restricted set can be executed extremely fast. We provide the pseudo code of our method in Algorithm 2.

The VAE-representation space can be replaced by other representations. Its advantage is in being derived via an unsupervised method which requires no labels, and can be performed only once, prior to label acquisition. The underlying assumption is that the representation space bears structure that correlates with the class function, and therefore can be used to restrict the pool set to a smaller, yet, impactful set of candidates for annotation.

### 4.1 Diffusion on graphs

Consider the optimized latent representation $g(D) = Z$, where $g : \mathbb{R}^d \to \mathbb{R}^k$, and $k$ is the dimensionality of the latent space $Z$. In this latent space we construct a weighted KNN proximity graph $G = (V, E)$, where the nodes $V$ correspond to the latent space representation of the data points $z_i \in Z$, and the edges $E$ corresponds to the pairs $(v_i, v_j)$ corresponding to $(z_i, z_j)$ who are neighboring in $Z$. The edge weights are computed as

$$W_{ij} = m\left(-\frac{\rho(\mathbf{z}_i, \mathbf{z}_j)}{\sigma_{ij}}\right) \mathbb{I}\{j \in N(i)\}$$

with $m : (z_i, z_j) \to \mathbb{R}_+$ as a similarity metric, $\rho$ as a distance metric, $\sigma_{ij}$ as a local scaling factor, and $N(i)$ as the $K$-NN neighbourhood of node $i$ in the latent space. We define a graph transition matrix by

$$M \doteq T^{-1}W, \tag{2}$$

where $T = diag(\sum_j W_{ij})$. $M$ stands for the transition probabilities of a Markov random walk on $G$.

we now construct a label diffusion framework as a Markov process to propagate the label information from the labelled set $D_l$ to the unlabelled set $D_u$, where The transition probability of a step from state $i$ to state $j$ is $M_{ij}$. W.l.g. consider a binary classification to facilitate the following: we associate the classification probability $P(\mathbf{y}(\mathbf{z}_i) = 1|\mathbf{z}_i)$ with a $t$-step hit probability - $P_t(\mathbf{y}(\mathbf{z}) = 1|i)$ of a random walk from $z_i$ to a

training sample $z$ with label 1. We can therefore predict the label '1' to $z_i$ in $Z_u$ based on the random walk probability $P_t(\mathbf{y}(\mathbf{z}) = 1|i)$. $p_t(y(x) = 1|i)$ can be derived by the recursive relation

$$p_t(y(x) = 1|i) = \sum_j p_{t-1}(y(x) = 1|j)p_{ij}. \tag{3}$$

Let $\chi_i = 2P_t(y(z) = 1|i) - 1 \in [-1, 1]$ denote an approximation to the binary label function of a node $v_i$. We also denote $\chi_u$ and $\chi_l$ as the entries corresponding to the unlabeled and labeled node sets in $\chi$, respectively. In matrix form we can rewrite (3) for $\chi$

$$\chi_u = [T_{uu}^{-1}W_{ul}|T_{uu}^{-1}W_{uu}]\begin{pmatrix} \chi_l \\ \chi_u \end{pmatrix}, \text{ where } T = \begin{pmatrix} T_{ll} & 0 \\ 0 & T_{uu} \end{pmatrix}, W = \begin{pmatrix} W_{ll} & W_{lu} \\ W_{ul} & W_{uu} \end{pmatrix}. \tag{4}$$

We can rewrite (4) the graph Laplacian $L = D - W$ in the system

$$L_{uu}\chi_u = W_{ul}\chi_l \iff L_{uu}\chi_{uu} = -L_{ul}\mathbf{y}_l \tag{5}$$

Equation (5) above can be solved via the iteration (Chapelle et al., 2009):

$$\chi_i^{(t+1)} = \frac{1}{L_{uu,ij}}\left(-(L_{ul}\mathbf{y}_l)_i - \sum_{j \neq i} L_{uu,ij}\chi_j^{(t)}\right), \tag{6}$$

where the superscript corresponds to the iteration index. Equation (6) is transducing, at time step $(t + 1)$, a label $\chi_i^{t+1}$ to $\chi_i$ as a weighted average of the labels of its neighbors at time step $t$.

Now consider the multi-class case, $\chi$ is now an $n \times C$ matrix. For the signal column vector corresponding to the class $c \in C$, the diffusion process initializes with

$$\chi_{ic}^{(0)} = \begin{cases} 1 & \text{if } \mathbf{z}_i \in Z_l \text{ and } c = \mathbf{y}_i \\ -1 & \text{if } \mathbf{z}_i \in Z_l \text{ and } c \neq \mathbf{y}_i \\ 0 & \text{if } \mathbf{z}_i \in Z_u. \end{cases} \tag{7}$$

The labels are propagated to $\chi_u$ gradually for $t$ steps. At the $(t + 1)$-th step,

$$\chi_{ic}^{(t+1)} = \begin{cases} 1 & \text{if } \mathbf{z}_i \in Z_l \text{ and } c = \mathbf{y}_i \\ -1 & \text{if } \mathbf{z}_i \in Z_l \text{ and } c \neq \mathbf{y}_i \\ (M\chi_{:,c}^t)_i & \text{if } \mathbf{z}_i \in Z_u. \end{cases} \tag{8}$$

## 4.2 Query criterion

To this end the diffusion process can be used to probe the most uncertain points. Most uncertain points maximize the gradient of the loss function in expectation and therefore will reduce the expected error in the model in equation 1. Active learning therefore, aims to query points of highest uncertainty.

The matrix $\chi^{(T)}$ of propagated values can be interpreted as **uncertainties** measured by the absolute value $|\chi_{c,i}^{(T)}|$. Specifically, the absolute value magnitude represents a measure of uncertainty on whether vertex $i$ belongs to class $c$. The magnitude can be used to select the new batch to query as

$$D'_{sub} = \{x_i : \arg\min_{\mathbf{z}_i \in \mathbf{Z}_u}^B \min_{c \in [C]} \|\chi_{c,i}^{(T)}\|\}, \tag{9}$$

where $\min^B$ denotes the $B$ smallest elements.

We demonstrate the exploration-refinement trade-off of (9) in Figure 3, with a subset of MNIST (LeCun et al., 1998) representation in a trained ResNet18 (He et al., 2016). Higher query rate is observed close to the decision boundary after the overall clusters have been explored (queried samples are in bold round points).

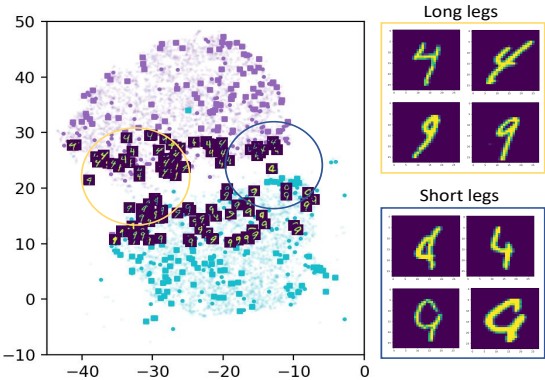

Figure 3: The DGMG selection of digits 4s and 9s (from MNIST) in an embedding space of ResNet18 well-trained with full data and labels.

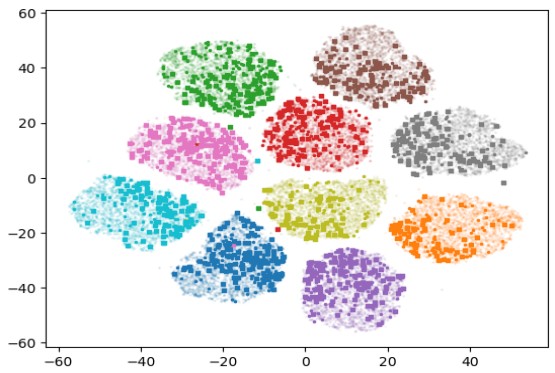

Figure 4: The DGMG selection of CIFAR10 in a ResNet18 embedding space. The dark dots are selected points by DGMG.

We also demonstrate that the query selection along the boundary captures '4's and '9's that have similar shape, 'leg' size, and orientation. These samples whose similarity can be best learned via annotation are automatically selected by our criterion (9) for annotation.

In Figure 4 plot the latent representation of CIFAR10 (Krizhevsky et al., 2009). We mark the points selected by the first five rounds as dark, round dots and data selected by the next five rounds as rectangular, dark dots. Different color stands for different classes and data in lighter color are the latent representation of all the CIFAR10 data. Here a similar trend is captured showing the sample selection start with exploration and then tends to the refinement of decision boundaries between clusters of different classes.

**Exploration and Refinement.** Our query criterion coupled with the diffusion process allows to explore the data set at early stages of active learning and switch to refinement when exploration has saturated. To understand this mechanism we show in the following that the diffusion iterant $\chi^t$ converges to the second eigenvector $\phi_2$ of the graph's Laplacian as $t \to \infty$. As we show below, asymptotically $\phi_2$ provides a relaxed solution to the minimal normalized cut problem, where the cut corresponds to the decision boundary between the two classes in $G(V, E, W)$.

At early stages of label acquisition low magnitude entries in $\chi$ correspond to data points that are unreachable from the training set via diffusion and need to be explored. At later stages all unlabelled data points $X_u$ are reachable via diffusion from the labelled set $X_l$. At this stage low mag-

---

**Algorithm 2** The DGAL strategy

**Input:** Labeled data $D_l$, unlabeled pool $D_u$, the initial VAE model $g$, the task model $f_\theta$, $T$, $K$, batch size $B$, round $R$, a query strategy $Q$
**Train** $g$ with the full dataset (without label information)
**Build** graph $G = (V, E, W)$ from the latent space $g(D) = Z$
**for** $r = 1$ **to** $R$ **do**
    Initialize $\mathcal{X}^{(0)}$ based on $D_l$
    **for** $t = 1$ **to** $T$ **do**
        $\mathcal{X}^{(t)} \leftarrow M\mathcal{X}^{(t-1)}$
        assign $\mathcal{X}^{(t)}$ with training labels
    **end for**
    **Query a restricted set** $D'_{sub} = \{x_q : \mathbf{z}_q = \arg\min_{i \in Z_u, c \in [C]} |\mathcal{X}^{(T)}_{i,c}|\}$
    **Active sub-sampling of batch $B$ from $D'_{sub}$ with criterion Q and network $f_\theta$:** $D^\star \leftarrow Q(f_\theta, D'_{sub})$
    **Annotate** $D^\star$
    **Update** $D_l \leftarrow D_l \bigcup D^\star$
    **Train** $f_\theta$ with $D_l$
**end for**

---

nitude entries correspond to the transition between the two classes -1 and 1. These nodes capture the eigenvector's transition from negative to positive entries. Therefore, sampling these points corresponds to the refinement of the decision boundary. We provide the main theoretical result on the convergence of $\chi^{(t)}$ and refer for further details in Kushnir & Venturi (2023).

**Lemma 4.1.** *Let $\lambda_1, ..., \lambda_n$, $\phi_1, ..., \phi_n$ be the solutions to the system: $L\phi = (1 - \lambda)D\phi$. Then $\chi^{(t)}$ converges to $\phi_2$ via the iteration (6) with $M$, as $t \rightarrow \infty$.*

### 4.3 Deep active learning with the task classifier

At the last stage of the algorithm we input the restricted subset $D'_{sub}$ into the task net and using a baseline active learning criterion we select the final set $D^*$ to be sent for annotation. Clearly feeding pool data into the network poses a significant computational cost. However, since the restricted set $D'_{sub}$ is significantly smaller than $D_u$ we observe a significant speedup in the query time. In our experiments we used two simple baseline to demonstrate the speed of DGAL. However, any other deep active learning criterion can be used, including another diffusion-based criterion (Kushnir & Venturi, 2023).

**Least Confidence.** Confidence sampling (Lewis, 1995): $x^* = \arg\max_x (1 - f_{\hat{c}}(x))$, where $f(x)$ is the prediction probability extracted from the output layer and $\hat{c} = \arg\max f(x)$. Denoted as **DGLC**. **Margin.** Margin sampling (Scheffer et al., 2001): $x^* = \arg\min_x (f_{\hat{c}_1}(x) - f_{\hat{c}_2}(x))$, where $\hat{c}_1, \hat{c}_2$ are the first and second most probable prediction labels respectively. Denoted as **DGMG**.

### 4.4 The VAE-DGAL algorithm

We provide VAE-DGAL pseudo code in Algorithm 2. We note that the input to VAE-DGAL includes the labeled and unlabeled pool set, the VAE architecture, and the task classification network $f_\theta$. After training the representation model $g$, we start the active learning cycles that entails diffusion based selection for restricting the pool set, feeding the restricted set to the task network $f_\theta$ and using a baseline active learning criterion to select the final set. After augmenting the training set with the final set, $f_\theta$ is trained again and the process repeats.

## 5 Experiments

As shown in this section, our experiments validate our goal achievement to reduce the query time while maintaining highest accuracy. We report DGAL improved query time vs. accuracy trade-off and compare it with pivotal SOTA baselines, including SEAL which aims, as well, to reduce query time. Additionally, we provide classical active learning empirical analysis results of the trade-off between the number of queried data points and accuracy, and we provide average query times, for all baselines and data sets. We also provide an ablation study with VAE-SEALS (see Algorithm 3) and with random versions of pool set restriction. Training details are provided in table 1 in the appendix.

### 5.1 Benchmarks

- **Random**: selects samples uniformly at random for annotation.

- **Confidence based methods**: A set of conventional selection strategies includes Least-Confidence (LC) (Lewis, 1995), Margin (Scheffer et al., 2001; Luo et al., 2005) and Entropy (Holub et al., 2008). In LC, the instance whose prediction is least confident is selected; Margin selects data that has minimum difference between the prediction probability of the first and second most confident class labels. Entropy selects samples that are most uncertain over all class probabilities in average.

- **BADGE** (Ash et al., 2019a;b): Selects points based on their gradient magnitude and diversity.

- **CoreSet** (Sener & Savarese, 2017a;b): An algorithm that selects a core-set of points that $\lambda$-cover the pool set.

- **SEALS** (Coleman et al., 2022a;b): restricts the candidate pool to the nearest neighbors of the currently labeled set to improve computational efficiency. We examine two versions of SEALS, one which updates the feature extractor, and another that sets it fixed using a VAE.

- **GANVAE** (Sinha et al., 2019a;b): A task-agnostic method that selects point that are not well represented in the pool, using a VAE and an adversarial network.

**Data sets and setting.** Experiments are conducted on multiple data sets to evaluate how DGAL performs in comparison to SOTA benchmarks. We also perform an ablation study. We experimented with benchmark data sets MNIST LeCun et al. (1998), EMNIST Cohen et al. (2017), SVHN Netzer et al. (2011), CIFAR10 Krizhevsky et al. (2009), CIFAR100 Krizhevsky et al. (2009), and Mini-ImageNet Ravi & Larochelle (2017) data sets. We include classic networks CNN LeCun et al. (1998), ResNet18 He et al. (2016), ResNet50 He et al. (2016), and ViT-Small Dosovitskiy et al. (2020).

## 5.2 DGMG vs. Benchmarks

**Test accuracy vs. query time.** In Figure 5 we report accuracy per total query time for a fixed number of queries. We observe for all data sets that DGMG's accuracy per query time is by far higher than SOTA. Moreover, we also emphasize its ability to reach the highest accuracy vs. the random baseline, which is the fastest method with essentially close to zero query time (approx. $10^{-3}$ secs.). It can be seen that the acceleration of DGAL is of order of x1000 with respect to the lowest performing benchmark method and x100 with respect to the next best performing method. These results emphasize the advantage of DGALs graph diffusion approach in selecting more impactful samples for the pool set restriction. In particular, the advantage over using the nearest-neighbors to the existing training set, as proposed in SEALS. The KNN criterion does not select a diversified set or even focus on decision boundaries refinement in the restricted set. Consequently, its accuracy gain in each query step is lower. Additionally, the ever growing candidates size in SEALS (by factor of $k$) is increasing the total query time.

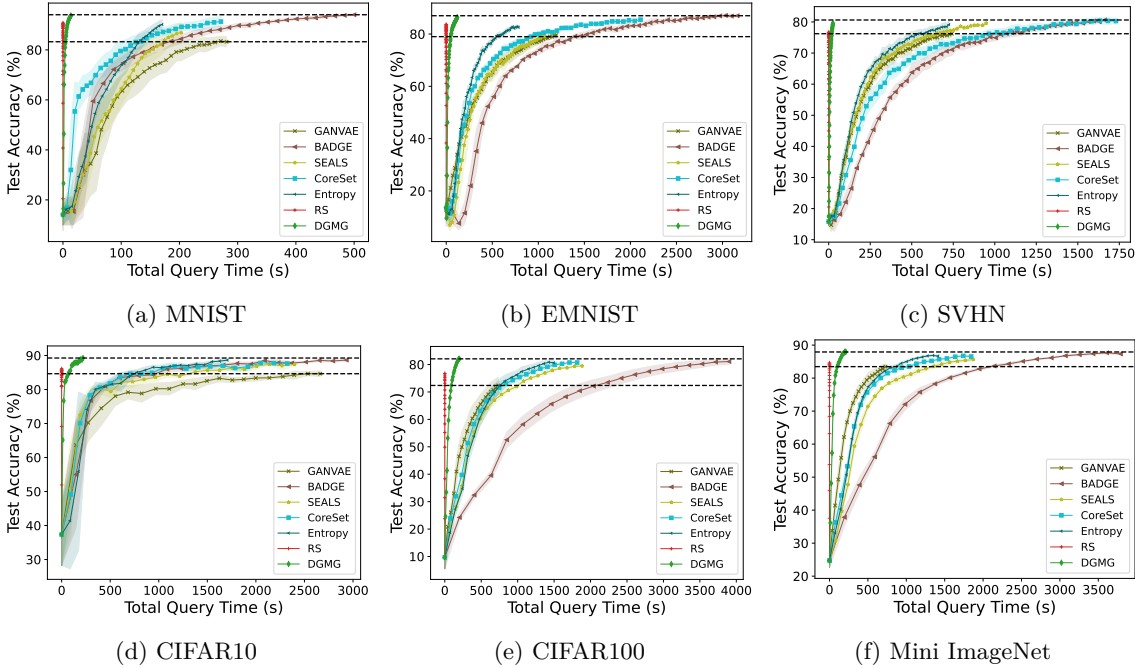

Figure 5: Plots of test accuracy vs. total query time for 6 datasets over 6 benchmarks. The horizontal lines capture the highest and lowest test accuracy among all methods.

We note that in our experiments for Figure 5, SEALS feature extractor is trained with labelled data prior to any acquisition cycles by using a part of the labelled data in order to implement the $k$-nearest neighbors (KNN) data structure. The remaining data serves for pool acquisition. Clearly, such an approach is not applicable to active learning where initial labels may not even exist, unless transferred from a pre-trained model, as suggested in Coleman et al. (2019). To provide a consistent benchmark, i.e. applying SEALS in the same way for all datasets, we adjust the algorithm by feeding the datasets into the trained task model to get an embedding representation as the feature extractor, which adds up query time. Below, we demonstrate another version of VAE-SEALS when using our unsupervised VAE for fixing the feature extractor to compute

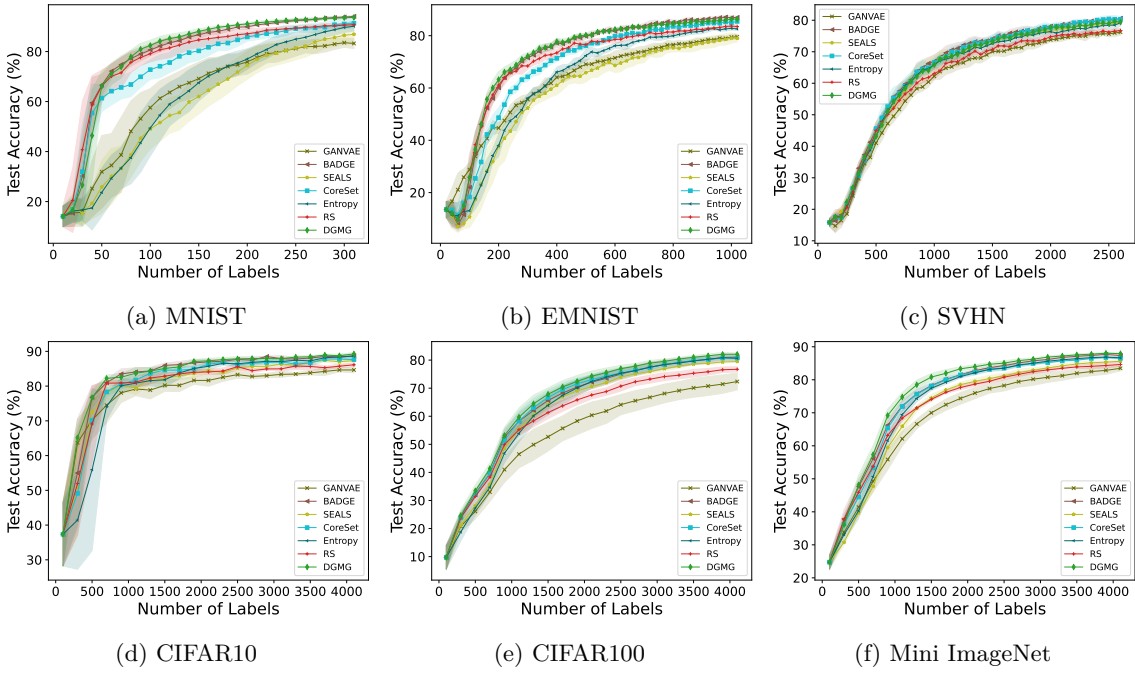

Figure 6: Plots of test accuracy vs. size of queried data. Each experiment is run at fixed query rounds for different methods and has been repeated 5 times.

the KNN graph only once, discarding the featurizer from the query time calculation. Even then SEALS is slower than DGAL in query time and worse in its query-vs-accuracy trade off.

**Test accuracy vs. number of selected labels.** In Figure 6, we report the classical active learning trade off between accuracy and training set size. DGAL is observed to be competitive with several SOTA benchmarks, in particular, BADGE of Ash et al. (2019a) and CoreSet of Sener & Savarese (2017a). SEALS on the other hand, is observed to be lagging in several data sets in early stage of the active learning because its selection criterion is relying on the nearest neighbors of the existing training set which does not diversify the restricted set enough. In fact, even its accuracy in the late stage of active learning does not get close to existing benchmarks because refinement is not occurring either in the nearest neighbor-based query criterion.

**Average query time.** In Figure 7 DGALs average query time is an order of magnitude lower than that of all benchmarks. Note that random (RS) baseline has an almost zero query time (at most $10^{-3}$ secs).

**VAE and graph construction time.** We provide running time for the VAE and graph construction in Table 2 in the appendix. As seen, even with the pre-processing time our method is still faster than other methods, in particular, faster than the SEALS algorithm **without** its additional pre-processing time. We note that the pre-processing time is a one-time procedure, while the query is repeated multitude times, depending on each case. Hence, total query time can increase significantly higher than the pre-processing time. Hence the advantage of reducing it even with the price of pre-processing time.

### 5.3 DGMG vs. VAE-SEALS in accuracy and query time

In this section, we compare DGMG and DGLC with VAE-SEALS (see Algorithm 3). In VAE-SEALS, instead of extracting the features of unlabeled data from the task model and re-computing its KNN graph at each round, we use the same VAE latent space as used in DGMG. We note that in the SEALS paper Coleman et al. (2019), the authors didn't specify the 'featurizer' or mention a specific representation space in the algorithm.

To some extent this is another ablation study to verify that our query component based on the diffusion is outperforming existing KNN criterion of SEALS, where the same representation is used. We provide Figure 8

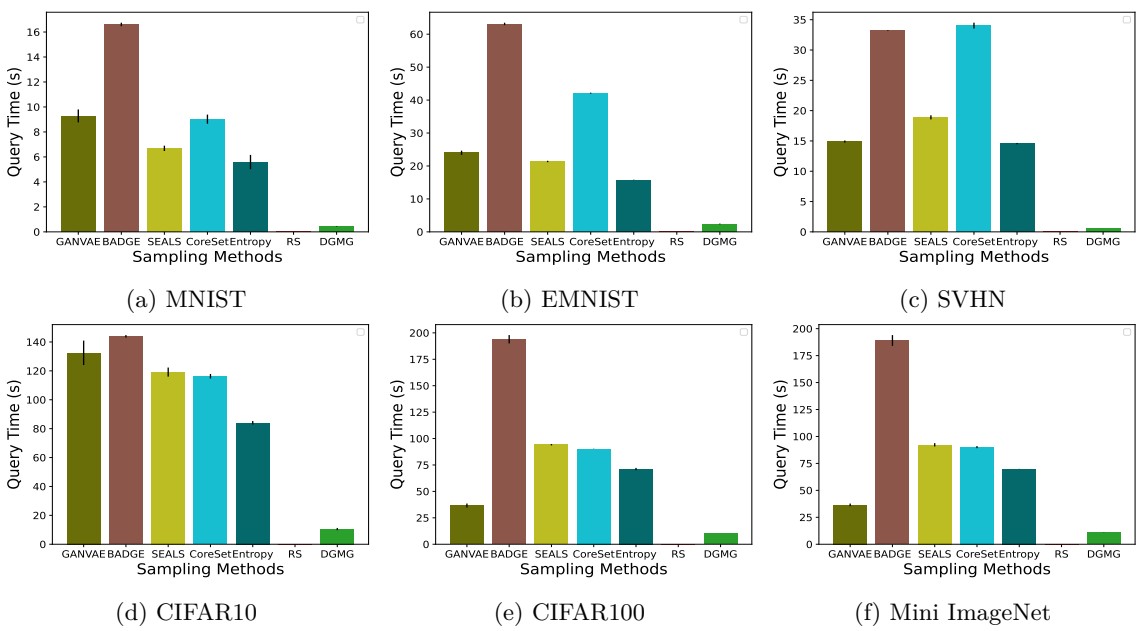

Figure 7: Query time for 6 benchmarks and DGMG, averaged over a fixed number of rounds for each method. The average time is averaged over 5 repetitions for each experiment.

below and figures 9 and 10 in the appendix. This study provides different experimentation on using a static pre-computed data representation with the KNN criterion, and its effect on the overall accuracy and query time of the methods. It also provides a view on the different versions of DGAL (DGMG and DGLC) and the advantage of the DGAL latent diffusion methodology over other restriction criteria. In particular, the comparison between VAE-SEALS and DGAL versions shows that DGAL is faster and achieve better accuracy. And, as seen in Figs. 8, DGMG has better accuracy than DGLC in most data sets.

In Figure 8, DGAL versions have an advantage over the VAE-SEALS. The VAE-SEALS can achieve similar accuracy with EMNIST, SVHN, CIFAR10 and Mini-ImageNet but uses more time than ours in these cases. It shows that the diffusion graph on latent space is more powerful than simply applying KNN on a latent space. On the other data sets SEALS doesn't achieve the accuracy that DGAL achieves within the same range of queries.

Comparing DGLC with VAE-SEALS in Figure 9, we observe the importance of pre-selection. The pre-selection of SEALS lacks diversity at the first few rounds of data acquisition. Under the same sub-sampling method (the least confidence sampling) after pre-selection, DGLC has an obvious advantage over VAE-SEALS at the beginning of the active learning cycles, especially for MNIST, EMNIST, CIFAR10 and CIFAR100. Also we see in this experiment that DGMG outperforms DGLC in accuracy over most data-sets. Mostly due to Margin's exploratory nature.

---

**Algorithm 3** The VAE-SEALS strategy

**Input:** Labeled data $D_l$, unlabeled pool $D_u$, the initial VAE model $g$, the task model $f_\theta$, batch size $B$, round $R$, a query strategy $Q$, label $y$
**train** $g$ with $D_l$
**implement** the $k$-nearest neighbors structure $\mathcal{N}(\cdot, \cdot)$ on the latent space $g(D) = Z$
**initialize** the limited unlabeled pool as
$D_u = \cup_{(Z,y) \in D_l} \mathcal{N}(Z, k)$
**for** $r = 1$ **to** $R$ **do**
    **active sub-sampling of a batch** $B$ **from** $D_u$ **with**
    **criterion Q and network** $f_\theta$**:** $D^\star = Q(f_\theta, D_u)$
    **annotate** $D^\star$
    **update** $D_l = D_l \bigcup D^\star$, $D_u = D_u \setminus D^\star) \cup N(D^\star, k)$

    **train** $f_\theta$ with $D_l$
**end for**

In Figure 10, we observe that in most cases, our
methods (DGMG, DGLC) win over the others in average query time. But still the overall difference is quite close. Due to the nature of the restriction process all methods reduce the query time due to a smaller pool set. Nevertheless, as we see in Figures 8 and 9, while VAE-SEALS average query time may be similar to DGAL's, its accuracy trade-off with the query time is significantly worse than DGALs.

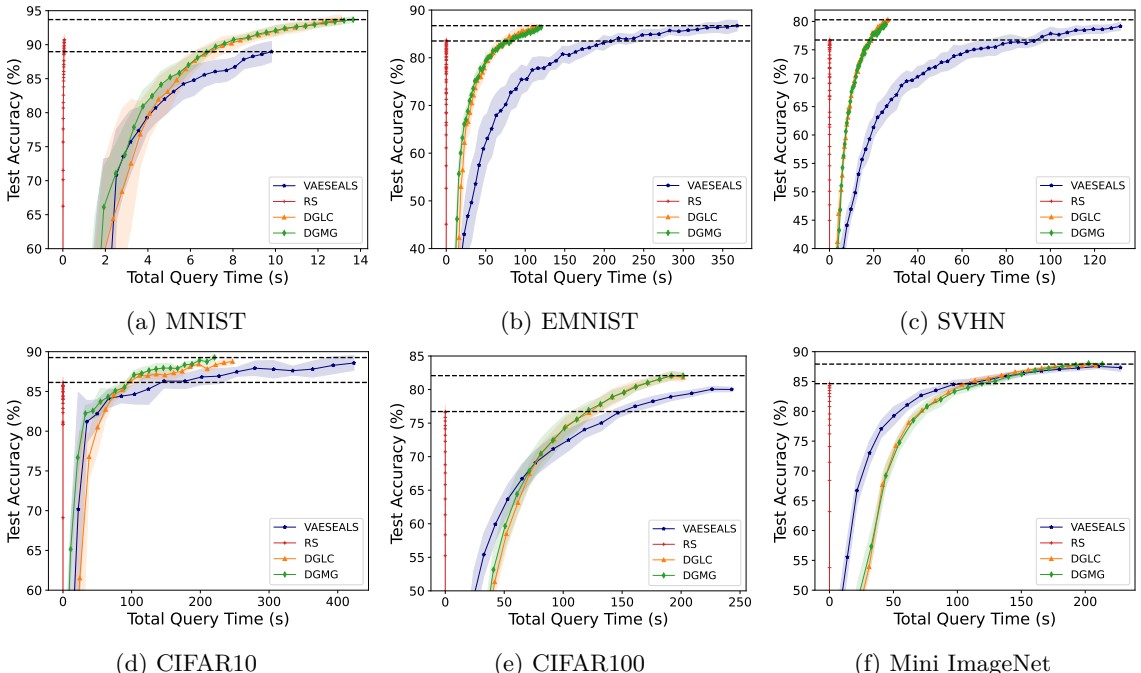

(a) MNIST        (b) EMNIST        (c) SVHN

(d) CIFAR10        (e) CIFAR100        (f) Mini ImageNet

Figure 8: Test accuracy vs. total query time for VAE-SEALS, DGMG, DGLC, and Random (RS). The horizontal lines capture the highest and lowest test accuracy among all methods.

## 5.4 Ablation study

In an ablation study, we compare DGMG with RSMG, Margin and the baseline Random Sampling (RS). In RSMG, the pre-selection is random sampling while in DGMG it's based on label diffusion. Due to space limitation we provide plotted results in the appendix.

In Figure 11 we plot the test accuracy vs. size of queried data, and observe DGMGs advantage at the early stage of learning. For example, in Figure 11d, DGMG is above RSMG and RS from 100 to 700 at the total number of labels. Similar trend exists also in Figure 11b, Figure 11c, and Figure 11e. It demonstrates how the latent diffusion graph pre-selects better than random sampling. In Mini-ImageNet, DGMG is close to RSMG, perhaps because larger data sets require more exploration and the random baseline is a good explorer criterion. Margin sometimes achieves worse results at early learning. Yet it achieves similar accuracy as in DGMG at later stages of learning as it essentially uses a similar criterion but without the restriction. Yet, Margin's query time is worse than DGMG and RSMG as seen consistently in Figure 12.

In Figure 12 we provide the average query time for each method. We observe advantage for DGMG and RSMG in query time in comparison with Margin, mostly due to the efficiency of the restriction method. DGMG uses about 1/8th time of Margin's while achieving better test accuracy. While RSMG uses a small amount of query time, DGMG, in most cases, trades better additional time for better accuracy, as seen in Figure 11.

We also perform an experiment in which we substitute various deep AL methods after our diffusion-based restriction of the candidate set in Figure 13.

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

## A    Experimental settings

In the choice of network architectures, we employed different deep neural architectures: CNN, ResNet18, and small vision transformers (ViTs), as presented in Table 1. If not specifically mentioned otherwise, we optimized the network for the cross-entropy loss with SGD Kiefer & Wolfowitz (1952); Sutskever et al. (2013) optimizer at the learning rate 1e-3, momentum 0.9, weight decay 5e-4 for non-pre-trained models. The Adam Kingma & Ba (2014) optimizer with learning rate 1e-3 is used for pre-trained models. For methods that include VAE training, we use Adam optimizer with 5e-4 learning rate by default. For all experiments we adopted an early stopping criterion for epochs iterations with patience 10. For all the methods that need a data embedding feature space (i.e. CoreSet, SEALS and BADGE) from the task model at each round, we added a linear layer at each network architecture of size 50 before the output layer, which are extracted by Pytorch Hooks. All the experiments were carried out on a NVIDIA RTX A6000. All the datasets and neural network information in the appendix follows the same details as showed in Table 1 in the main text.

Data and architecture details are provided in table Table 1.

Table 1: A summary of data and experimental settings that we used in our paper. Here 'Pre_n' is the size of restricted subset $D'_{sub}$ in Algorithm 2. 'Budget' is the number of queried data points per round, i.e., the size of $D^\star$. 'Round' stands for the total sampling rounds.

| Dataset | Pool size | Label size | Input | Initial # of data | DimVAE | Pre_n/Budget /Round | Architectures | Pre-trained | Test size |
|---|---|---|---|---|---|---|---|---|---|
| MNIST | 60,000 | 10 | 28x28 | 10 | 50 | 100/10/30 | CNN | False | 10,000 |
| EMNIST letter | 124,800 | 26 | 28x28 | 20 | 256 | 200/20/50 | ResNet18 | True | 20,800 |
| SVHN | 73,257 | 10 | 32x32 | 100 | 100 | 500/50/50 | ResNet50 | False | 26,032 |
| CIFAR10 | 50,000 | 10 | 32x32 | 100 | 100 | 5k/200/20 | ResNet50 | True | 10,000 |
| CIFAR100 | 50,000 | 100 | 32x32 | 100 | 100 | 5k/200/20 | ViT-Small | True | 10,000 |
| Mini-ImageNet | 48,000 | 100 | 84x84 | 100 | 100 | 5k/200/20 | ViT-Small | True | 10,000 |

**Latent space generation.**    To get a representative latent space from data, we use a ResNet18 based encoder for CIFAR100 and MiniImageNet and a CNN-based VAE for the rest of the datasets. We use an Adam optimizer with learning rate 5e-4 and epoch 300. An early stopping criterion is used with patience 20. For the training of GANVAE, we use the same VAE and discriminator with learning rate 1e-4, total epochs 100 and no early stopping.

## B    Running time and parameter selection

The running time analysis is composed of three parts. The first part includes the computation of the $K$-NN graph. The $K$-NN proximity search computational cost can be reduced from the naive search cost by using procedures for $K$-NN search based on KD (Bentley, 1975) or ball trees (Omohundro, 1989). Such methods have complexity $O(dN \log N)$, with $d$ as the dimension of the space. Other alternatives include approximate search ((Datar, 2004)). In the second part, the diffusion vector $\chi$ is multiplied with the transition matrix. Addressing its sparsity as $O(KN)$ non-zero entries, this operation scales linearly in $N$ as $O(TKN)$. $T$ and $K$ determine the level of confidence imposed by the diffused training set over the unlabeled set. Higher $K$ imposes strong confidence in the current labeling hypothesis, but renders the diffusion more exhaustive. Similarly, large number of iterations $T$ may result in an overly smoothed (and less informative) signal $\chi^{(T)}$. During exploration, large $T$ imposes an hypothesis that may be locally correct but is far from being globally reliable. In our experiments, we use $T \simeq \log_K N$, with the intention to cover most of the graph via diffusion: assuming a diameter-balanced graph (S. Miklavic, 2018). The cover requires $O(\log_K N)$ iterations, if the labelled set is small (i.e. $|X_l| \approx O(1)$, as typical in active learning settings). We use $K$ sufficiently high to allow graph connectivity. This parameter selection leads to a diffusion process that scales as $O(KN \log_K N)$.

Finally, a batch of smallest soft-labels needs to be queried. This requires a quick-sort to be applied to the soft labels magnitude, which scales $O(N \log N)$. We conclude that the running time of DDAL is $O(dN \log N + KN \log_K N + N \log N)$, whereas for a constant $K$ can be further simplified to $O(dN \log N)$. We note that the number of units in the penultimate layer $d$ is typically small.

**Additional parameters:** The batch size $B$ is set at no more than 500 (see batch size for diffusion algorithm in Kushnir & Venturi (2023). The number of epochs are set based on a stopping criteria for the convergence of the loss function.

## C   Variants of multi-class extension

We propose in the following other possible formulations to extend the diffusion-based active learning criterion to the multi-class setting..

**One-vs-all approach**   In the one-vs-all setting the batch is queried according to

$$\hat{X} = \arg\min_{i \in X_u}^B q\left(\chi_{:,i}^{(T)}\right)$$

for some chosen function $q$ which measures a notion of uncertainty at point $x_i$, given the matrix $\chi^{(T)}$. In Section 4.2, we chose $q$ to be

$$q\left(\chi_{:,i}^{(T)}\right) = \min_{c \in [C]} \left|\chi_{c,i}^{(T)}\right|, \ or \ q\left(\chi_{:,i}^{(T)}\right) = \left\|\chi_{:,i}^{(T)}\right\|_p$$

for some $p \in [1, \infty]$.

**Multivariate diffusion approach**   Moving from the one-vs-all approach, we can performs the query as follows, using the property that $M$ is a stochastic matrix. For each data point $x_i$, we propagate a probability vector $\chi_i^{(t)} \in \Delta_C$. This vector can be initialized as

$$\chi_{i,c}^{(0)} = \begin{cases} 1 & \text{if } i \in \mathcal{X}_\ell \text{ and } c = y_i \\ 0 & \text{if } i \in \mathcal{X}_\ell \text{ and } c \neq y_i \\ \frac{1}{C} & \text{otherwise} \end{cases}$$

We can therefore diffuse the matrix aggregating the signal for all the points, $\chi^{(t)} \in \mathbb{R}^{N \times C}$, and diffuse it as in the binary case:

$$\chi^{(t)} = M\chi^{(t-1)}, \qquad \chi^{(t)}|_{\mathcal{X}_\ell} = \chi^{(0)}|_{\mathcal{X}_\ell}$$

Since $M$ is stochastic, it holds that $\chi^{(t)} \in \Delta_C$ at each iteration $t$. Therefore we can interpret each vector $\chi_i^{(t)}$ as a probability vector of the data point $x_i$ belonging to different classes, obtained by the diffusion above. It therefore makes sense to choose the points to query as

$$\hat{X} = \arg\min_{k \in \mathcal{X}_u}^B q\left(\chi_i^{(T)}\right)$$

where $q : \Delta_C \to \mathbb{R}$ is some measure of uncertainty. Possible choices include:

- Uncertainty: $q(p) = p_{c^*}$, where $c^* = \arg\max_c p_c$;

- Margin: $q(p) = p_{c^*} - p_{c_2^*}$, where $c^*$ is defined as above and $c_2^* = \arg\max_{c \in [C] \setminus \{c^*\}} p_c$;

- Negative entropy: $q(p) = \sum_{c \in [C]} p_c \log p_c$.

## D   VAE training and graph building time

Here we provide the total training time for VAE and graph construction time with latent variables.

## E   Supplamentary results

### E.0.1   DGMG vs. VAE-SEALS in accuracy and query time

Table 2: A summary of the VAE training time and graph building time for DGAL (measured in seconds).

| Dataset | VAE training | Graph construction |
|---|---|---|
| MNIST | 182 | 139 |
| EMNIST letter | 478 | 731 |
| SVHN | 276 | 207 |
| CIFAR10 | 208 | 108 |
| CIFAR100 | 1134 | 125 |
| MiniImageNet | 864 | 133 |

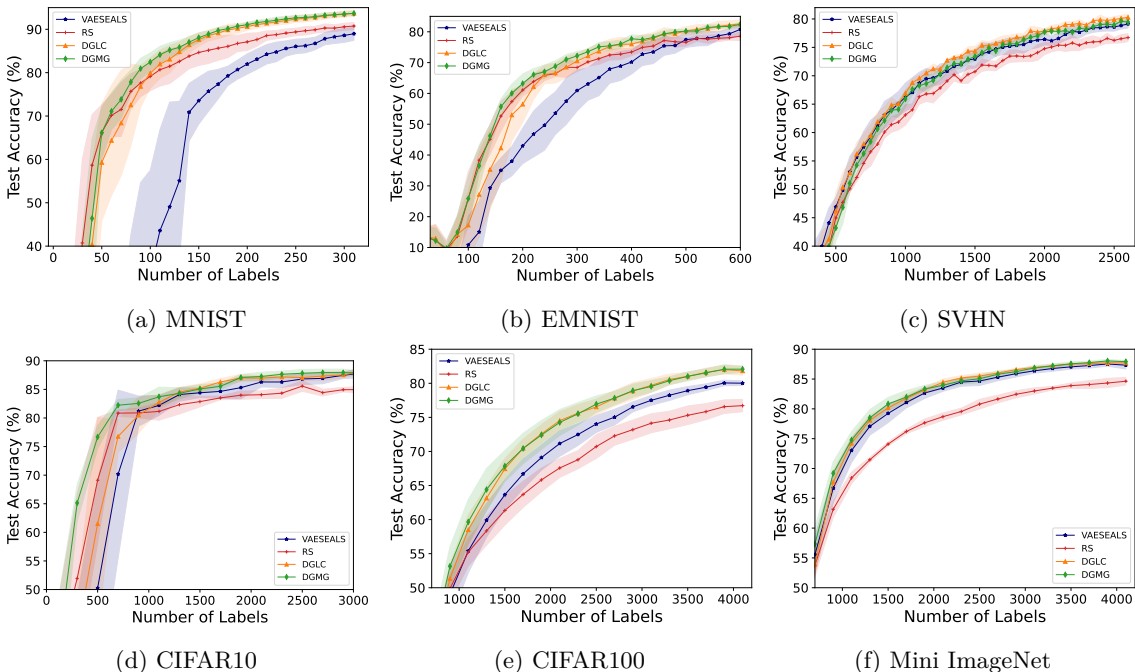

(a) MNIST           (b) EMNIST           (c) SVHN

(d) CIFAR10           (e) CIFAR100           (f) Mini ImageNet

Figure 9: Test accuracy vs. size of queried data for VAE-SEALS, DGMG, DGLC, and Random Sampling (RS) over limited total number of labels. Each experiment is run at fixed query rounds for different methods and repeated 5 times.

### E.0.2 Ablation study

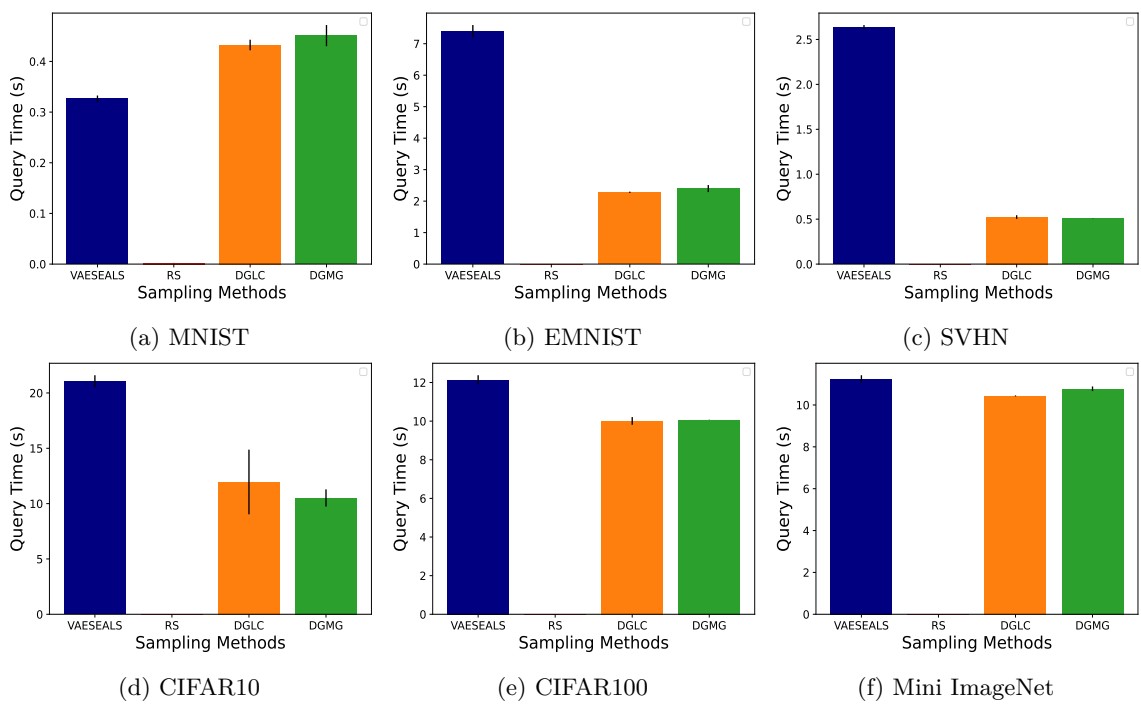

Figure 10: Query time for DGAL methods, VAE-SEALS and Random Sampling (RS), averaged over a fixed number of rounds for each method. The average time is averaged over 5 repetitions for each experiment.

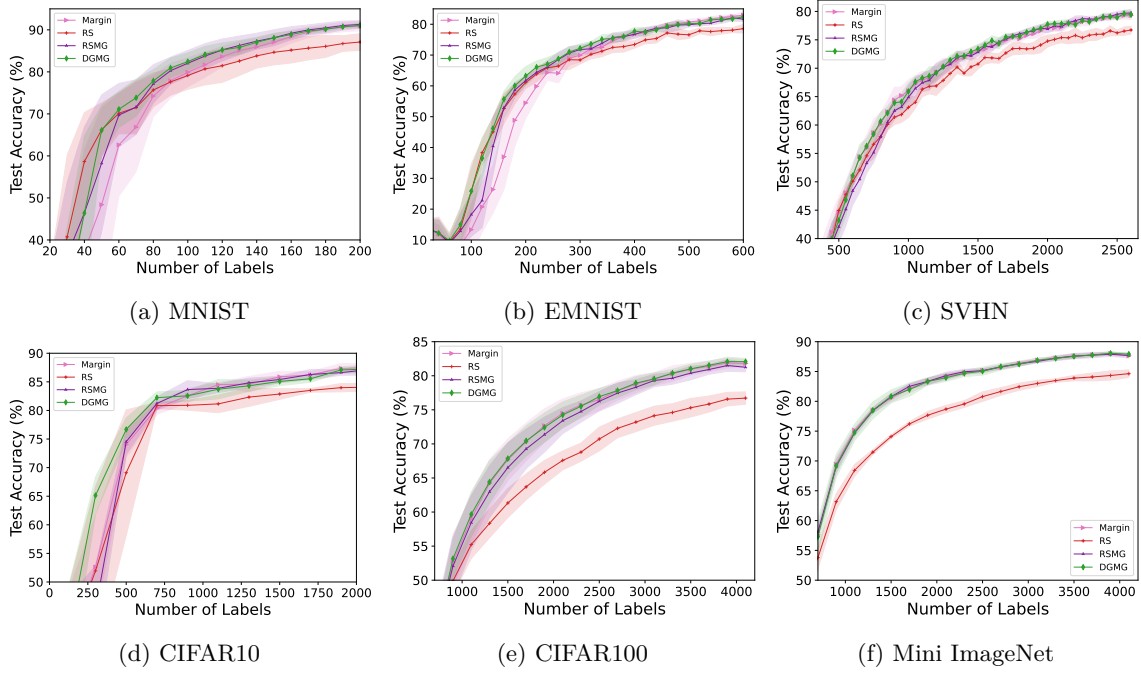

Figure 11: Test accuracy vs. size of queried data for 3 baselines and DGMG. Each experiment is run at fixed query rounds for different methods and has been repeated 5 times.

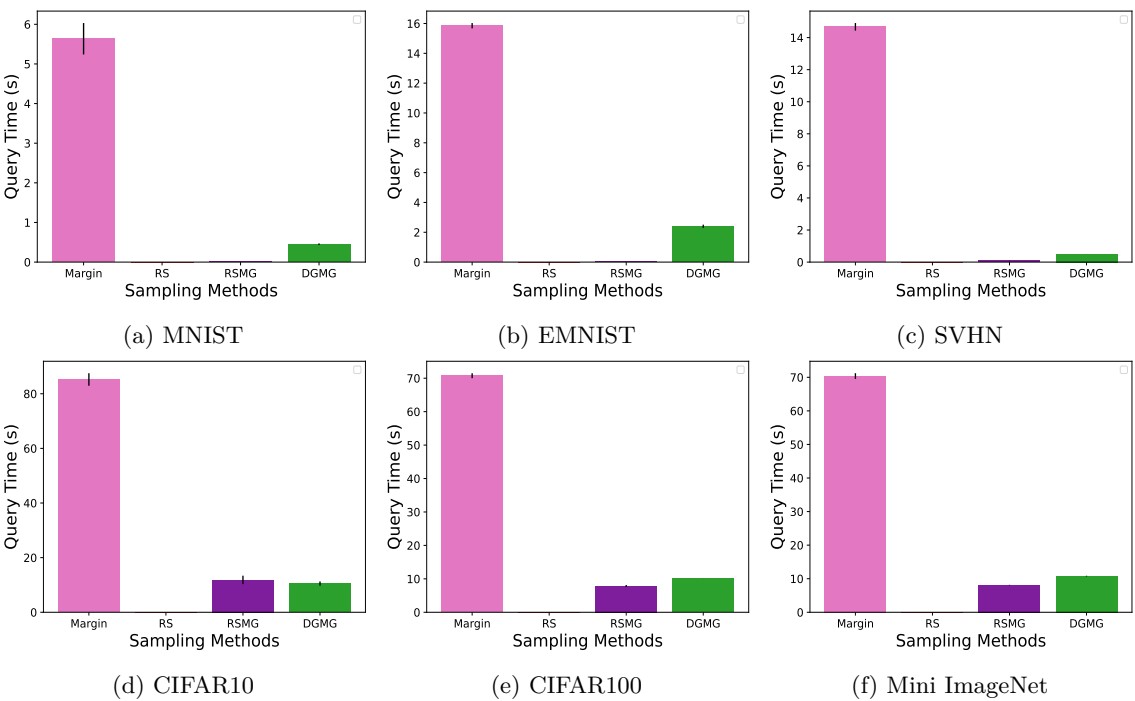

Figure 12: Query time for 3 baselines and DGMG, averaged over fixed rounds for each method. The average time is averaged over 5 repetitions for each experiment.

### E.0.3 Ablation study of DGAL with different deep AL criteria

We provide results of using additional AL criteria in the deep net architecture to test the acceleration based on the VAE and graph diffusion-based restriction.

It can be seen that DGBADGE may reach accuracy similar to DGMG but the trade-off with query time still renders it slower even on the restricted set in some of the tested data sets. For DGCoreSet the situation is even worse as we can see slower query time trade off with accuracy consistently for all data sets.

As seen deep net AL criterion that are fast to compute and accurate enough are still showing the best tradeoff. this is porbably due to the informative selection in the graph diffusion AL. Nevertheless, all methods combined with VAE diffusion active learning are clearly faster than running the active learning method alone, as seen in Fig. 5.

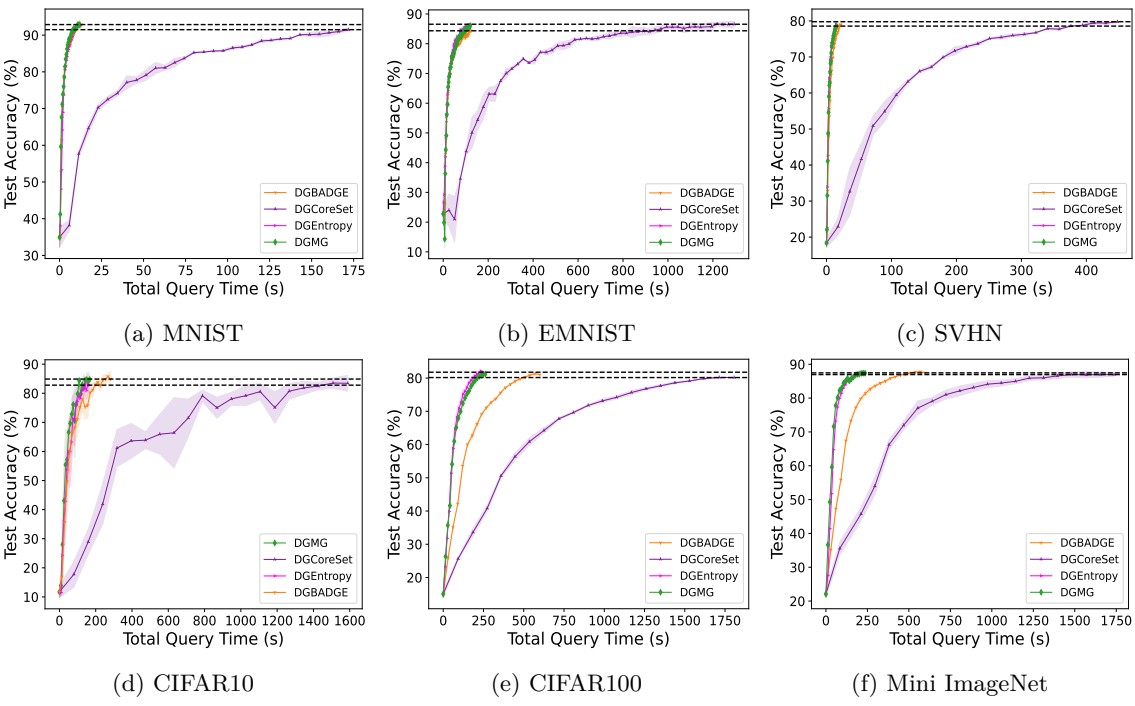

(a) MNIST  (b) EMNIST  (c) SVHN

(d) CIFAR10  (e) CIFAR100  (f) Mini ImageNet

Figure 13: Query time for 3 baselines and DGMG, averaged over fixed rounds for each method. The average time is averaged over 5 repetitions for each experiment.

