# OpenReview forum: "Accelerated Deep Active Learning with Graph-based Sub-Sampling"
_TMLR — Rejected by TMLR_

### Review · Reviewer_JxW6 · 2023-10-30

**Summary Of Contributions:**

The paper introduces Diffusion Graph Active Learning (DGAL), a two-stage active learning approach aimed at enhancing the efficiency of deep active learning methodologies. The fundamental concepts can be summarized as follows:

1. Utilize a Variational Autoencoder (VAE) to represent the data in a latent space.

2. Construct a graph of data samples based on this latent representation.

3. Employ graph diffusion techniques to select a narrow set of query candidates from the unlabeled data pool.

4. Feed these candidates to a specialized task model to finalize the query set using conventional active learning criteria such as uncertainty sampling.

The initial VAE representation and graph are computed once, while the candidate selection via graph diffusion operates in linear time. Experimental results over 6 image datasets indicate that DGAL attains a query wall-time speedup of 100x relative to conventional methods, all while maintaining competitive levels of accuracy.

**Audience:**

Yes

**Claims And Evidence:**

Yes

**Requested Changes:**

Please see Weaknesses above.

**Note**: I am not an expert on active learning or graph learning, so I do not know if this proposed paradigm is novel or useful for the active learning community. However, I do think that the paradigm of "unsupervised representation learning -> graph diffusion --> active learning" is interesting, and perhaps also useful to domains other than image data models (e.g., language model fine-tuning with actively selected samples).

**Strengths And Weaknesses:**

## Strengths
+ [**Better Computational Efficiency of Active Learning**]: The paper addresses a critical bottleneck in the field of active learning—the computational expense and time associated with querying extensive unlabeled datasets—which has thus far restricted the wider adoption of active learning in handling large datasets.

+ [**Sound Solution**] DGAL reduces the query time by applying VAEs for representation learning, graph construction for capturing structure, and graph diffusion for efficient querying. The methodology is intuitive and technically sound. Building a graph on the VAE latent space and diffusing labels is an elegant approach to select diverse and uncertain candidates.

+ [**Experimental Evidence**]: Comprehensive experimental validations are performed on six image datasets, encompassing various network architectures. The outcomes compellingly substantiate DGAL's capability to significantly expedite query wall-time, while maintaining or even surpassing the accuracy of existing state-of-the-art active learning strategies. The paper’s empirical results corroborate the utility of DGAL in accelerating the active learning process for deep neural networks without compromising accuracy. These advantages make DGAL a promising candidate for wider applications in the active learning community.

## Weaknesses

+ [**Other Representation Learning Methods**] The paper would benefit from additional ablation studies examining the impact of using VAE-based latent space against other representation learning methods, e.g., GAN, Self-Supervised Learning, CLIP. Such studies could further substantiate the choice of VAEs in the framework.

+ [**Effects of Hyperparameters**] DGAL involves several key hyperparameters, such as the number of diffusion steps (T) and the number of nearest neighbors (k) for kNN graph construction. A more detailed investigation into how these parameters influence the trade-off between accuracy and efficiency would be insightful. Especially, I care if the hyperparameter configuration is sensitive to the dataset choice. If so, this would make the method less applicable to real-world practices (an exhaustive search of hyper-parameters for each dataset would be a nightmare).

+ [**Sampling**] The paper employs basic uncertainty sampling techniques for the final query selection from the candidate subset. Consideration of more sophisticated active learning criteria for the second step could potentially enhance the model’s accuracy.

---

> ### Author Response · Authors · 2023-11-25
> **Addressing all comments and suggestions**
>
> We thank the reviewer for her\his thorough review, helpful comments, and positive view of our paper.\
>
> **[Effects pf Hyperparameters]**: Following the reviewers’ question we added section B in our appendix (due to space limitations) on parameter selection in our algorithmic setting. Our analysis demonstrates that the selection of key parameters has no dependence on the data set itself, under weak assumptions. Other parameters are set based on prior experiments and references.
>
> **[Sampling]**: Following the reviewer's comment we provide in our revision appendix E.0.3 with new results of using additional AL criteria in the deep net architecture to test the acceleration based on the VAE and graph diffusion-based restriction. We used the following methods with our VAE-based graph-diffusion method: DGBADGE, DGCoreSet, and DGEnotropy and compared them with our DGMG. BADGE and CoreSet are well know sophisticated active learning methods that, however, take some more computation time on each query step, but may yield higher accuracy.
>
> In our experiments we see that indeed DGBADGE is must faster than BADGE, and similarly for DGCoreSet and DGEntropy. Clearly this is a result of using our restriction method on the VAE graph with our diffusion based AL. DGBADGE is comparable in its accuracy-to-time trade-off to our DGMG, and also to DGEntropy, which are faster criteria to compute.
> For DGCoreSet the situation is worse as we can see slower query time trade off with accuracy consistently for all data sets. CoreSet computation is so high that even for the small restricted set it lingers over BADGE and the other more simple criteria Margin, and Entropy. Yet, DGCoreSet is faster than CoreSet.
>
> **[Other representation learning Methods]**: we agree that the question of whether the VAE representation methods yields better representation than others is an interesting one. In our paper, we do not claim that VAE is necessarily the best representation of choice, and in fact we don’t expect it to be such for every data set. We also do not consider the VAE choice as the main contribution of our paper. The main contribution of our paper is to propose an architecture in which a representation space can be constructed in an unsupervised manner and used efficiently to build a graph representation that can be used to restrict a large data set from being wholly fed in the deep net. Our other main contribution is also in proposing the diffusion based method in the restriction step, which leverages an optimal exploration-exploitation framework for query sampling.
>
>  The main intuition behind VAEs is that optimizing the reconstruction loss will yield good class representation IF the structure in the data points corresponds to the class assignment. This setting, we believe, is typical in imagery data. It may be the case that other unsupervised representations capture class structure as well, even though their loss function is not the reconstruction one.

---

### Review · Reviewer_JQe2 · 2023-11-01

**Summary Of Contributions:**

This paper proposed a two-step method for active learning which uses a VAE model to narrow down the search space and combines a graph diffusion to select the training examples to be labeled.
The proposed method was tested on image classification datasets.

**Audience:**

Yes

**Broader Impact Concerns:**

The author did not discuss the limitations or broader impact of this work.

**Claims And Evidence:**

No

**Requested Changes:**

- Please find appropriate references for the first four sentences.
- Instead of "dependant linearly on $k$", please write "dependent linearly on the number of neighbors $k$", which is more readable.
- Grammar: Let the graph laplacian $L = D - W$ in the system
- Using $=$ for assignment in algorithms is not a good idea.
- Why is VAAL renamed to GANVAE?

**Strengths And Weaknesses:**

## Strengths

- The idea of using a VAE to generate the similarity graph is reasonable.
- The motivation was clearly explained. The illustration was easy to understand.

## Weaknesses

- The author should improve the mathematical notation. The current presentation is unclear and confusing. For example:
  - The first line of math is already confusing. The author first defined $f_\theta$ to be a classifier without specifying its domain and codomain.
  - $\mathcal{D}$ was not defined.
  - The meaning of $[C]$ is not explained.
  - $\mathcal{P}_\mathcal{Z}$ is not defined.
  - The notation "a set $\sim$ a distribution (I assume)" is wrong.
  - A subset $D^\star$ of what set?
  - $D^\star$ does not appear in the first unnumbered equation.
  - The solution to this optimization problem is an empty set.
  - What is $g$ and $Z$ in $g(D) = Z$?
  - What are the nodes $V$ and edges $E$ of the graph $G$ (if the author follows the convention)?
  - If $m$ is a function, its domain and codomain were not specified.
  - What is $P_t$?
  - What are $\mathcal{X}\_i$, $\mathcal{X}\_l$, $\mathcal{X}\_u$, $\mathcal{X}\_{uu}$, $\mathcal{X}\_j^{(t)}$, and $\mathcal{X}\_{ic}^{0}$?
  - What is the $\mathcal{X}^{(t-1)} | D_l$ operation?
  - What is the range of $\arg\max f(x)$?

Generally, the author should significantly improve the clarity and readability.
I cannot assess the technical contributions of this paper based on its current form.

---

> ### Author Response · Authors · 2023-11-25
> **Addressing all comments and suggestions**
>
> We thank the reviewer for his\her thorough review and very helpful suggestions.
> We addressed in our revision every single remark and question that the reviewer has pointed to, without exception. We hope this would facilitate his\hers review.
>
> **Requested Changes**
> 1. **Please find appropriate references for the first four sentences.**\
> Done
> 2. **Instead of "dependent linearly on", please write "dependent linearly on the number of neighbors ", which is more readable.**\
> Done
> 3. **Grammar: Let the graph laplacian $L=D-W$ in the system Using for assignment in algorithms is not a good idea.**\
> Done
> 4. **Why is VAAL renamed to GANVAE?**\
> GANVAE is referred to as such to emphasize its use of the adversarial network in the paper we cited.

---

### Review · Reviewer_6DzJ · 2023-11-12

**Summary Of Contributions:**

This paper proposes a new way to accelerate active learning by using the graph diffusion process to restrict a small sub-set of unlabeled data before using some criteria to further select a set for annotating the labels. Additionally, all the processes are carried out on the latent space of a VAE.

**Audience:**

Yes

**Broader Impact Concerns:**

There are no broader impact concerns.

**Claims And Evidence:**

No

**Requested Changes:**

I suggest the following changes

1) Move Section 4.1 about VAE to the background section.

2) Significantly rewrite Section 4.2 because currently it is not understandable to me. The authors should provide the intuitions/motivations that link the graph diffusion process and active learning. How do you transition from binary classification to multi-class classification? How do you store label vector in Eq. (2) (i.e., vector of indices)? What is the meaning of $\mathcal{X}_i$ which seems to be a vector of $C$ dimensions?  Moreover, the motivation/explanation of how the objective function in Section 3 links to the proposed approach in Section 4 is also necessary.

**Strengths And Weaknesses:**

Strengths

- The idea of using the graph diffusion process in active learning is interesting.

Weaknesses
-  The paper is disorganized and the writing needs a significant improvement.
-  Most of technical details in Section 4.2 is almost not understandable with missing notions (e.g., matrix D and matrix T (is it a diagonal matrix?)), lack of explanations and motivations, ambiguous clarifications and explanations, and poor mathematical rigorousness.
-  There is no connection between the objective function in Section 3 (i.e., choosing the sub-set with smallest errors) and the methodology in Section 4. Indeed, due to the proposed objective function, we only need to query the misclassified points.

---

> ### Author Response · Authors · 2023-11-25
> **Addressing all comments and suggestions**
>
> We thank the reviewer for his\hers review and thoughtful comments. We  addressed all of reviewer's 6DzJ items in the revision, and below:\
> **Weaknesses:**
> 1. **The paper is disorganized and the writing needs a significant improvement.** We have taken this remark with our revision and executed changes as the reviewer suggested: we revised section 4.2, added definitions, rigor, additional equations, and we moved the discussion of VAE to the background.
>
>  2. **Most of technical details in Section 4.2 is almost not understandable with missing notions (e.g., matrix D and matrix T(is it a diagonal matrix?)),**. We revised section 4.2 (revised as 4.1) and updated the mathematical notation, definitions, motivations, and rigor to address this comment as well as reviewer's jQe2 specific corrections.
>
>  3. **There is no connection between the objective function in Section 3 (i.e., choosing the sub-set with smallest errors) and the methodology in Section 4. Indeed, due to the proposed objective function, we only need to query the misclassified points.**
> We edited the text and added the connection to section 4.2, please check the text within.
> We would like to further clarify this fundamental point in active learning based on uncertainty sampling. Since one cannot query the points that are miss-classified  (because we simply don't know their ground-truth), active learning  selects points that potentially minimize the error. Uncertainty sampling selects the points that the model is most uncertain about. Therefore for such points the model is more prone to make mistakes on them. Querying such points clears the uncertainty and the expected error.
> To further elaborate on this result consider
> $\nabla_{\theta_{n+1}} \mathcal{L}(f_\theta(x), y) = \nabla_{\theta_{n+1}} f_\theta(x)\cdot \frac{\partial \mathcal{L}(f, y)}{\partial f} =h^{1:n}(x) \cdot f_\theta(x) (1 - f_{\theta(x)}) \cdot (\frac{f_\theta(x) - y}{f_\theta(x)(1-f_\theta(x))})= h^{1:n}(x)\cdot (f_\theta(x) - y)$, where $h^{1:n}(x)$ is $h^i_{\theta_i} : R^{d_{i-1}} \to R^{d_{i}}$ denote the inner layers of the network (with $d_0 = d$).
>
> This expression implies that $\|\nabla_{\theta_{n+1}} \mathcal{L}(f_\theta(x), y)\|$ is large when $f_\theta(x)$ is far from the true label $y$.
>  In expectation this is maximized for the most uncertain points.
> In our deep learning case case we probe uncertainty via a diffusion process on a graph constructed from the penultimate representation. The magnitude of the diffused labels is an indicator of the level of uncertainty the model has about them.
>
> **Requested Changes: I suggest the following changes**
> 1. **Move Section 4.1 about VAE to the background section.**\
> Done
> 2. **Significantly rewrite Section 4.2 because currently it is not understandable to me.**\
> Done
> 3. **The authors should provide the intuitions/motivations that link the graph diffusion process and active learning.**\
> Done
> 4. **How do you transition from binary classification to multi-class classification?**\
>  This is covered in equations 4 and 5 as well as in appendix B. for the multiclass case $\chi_i$ becomes a matrix of columns corresponding to a class assignment and rows correspond to an index of a point.
> 5. **How do you store label vector in Eq. (2) (i.e., vector of indices)?**\
> If we understand the question correctly, this is a python array and every entry of the label vector corresponds to a soft label assignment that is inferred during the graph diffusion process
> 6. **What is the meaning of $\chi_i$ which seems to be a vector of $C$ dimensions?**\
> In the multiclass case (i.e. more than 2 classes) $\chi_i$ is a matrix with columns corresponding to classes and rows corresponding to data points. I.e. Each entry $(i,c)$ corresponds to the assignment of point $i$ to a class $c \in C$.
> 7.  **The motivation/explanation of how the objective function in Section 3 links to the proposed approach in Section 4 is also necessary.**\
> We have introduced that change in the revised text.

---

### Author Response · Authors · 2023-11-25
**Revision and comments. Thank you for your review and helpful comments**

We thank all the reviewers for their insightful comments which helped to improve our manuscript. We uploaded a revision to reflect all changes, and answered the reviewers questions and comments below.

Thank you,
the Authors

---

### Decision · Action_Editor_yfTz · 2023-12-21

**Recommendation:** Reject

**Comment:**

As discussed above, the biggest concern raised by the reviewers was that the notation and definitions in the paper were unclear. While the authors provided an updated draft of the paper, the original paper was sufficiently hard to understand that the reviewers were unable to effectively review the paper. The authors did justify their choice of the VAE in the rebuttal as being not critical to the method, but this should be made clear in the paper. I think it's possible that the paper could be fit for publication after substantial revisions/rewriting for clarity.

**Audience:**

Yes.

**Claims And Evidence:**

Reviewers raised concerns about the experimental setting used to test the proposed method (including a lack of baselines). More broadly, reviewers shared many concerns about the paper having notational issues and being hard to read/follow. I'm not sure whether that counts as "unclear evidence", but if the definition of the method is unclear, it is unlikely that the method will be impactful.

**Resubmission Of Major Revision:**

The authors may consider submitting a major revision at a later time.

---

> ### Author Response · Authors · 2024-01-08
> **Commentary and additional correction for resubmission**
>
> We thank the chair for his commentary and would like to ask for post-rebuttal additional comments made for clarity/notation beyond the multiple corrections we already applied.   We would like to apply them towards the suggested resubmission.
>
> Also, we would like to draw the action-editor's attention to the fact that the VAE justification actually has been addressed in the paper's revision in section 4:
> “The VAE-representation space can be replaced by other representations. Its advantage is in being derived via an unsupervised method which requires no labels, and can be performed only once, prior to label acquisition. The underlying assumption is that the representation space bears structure that correlates with the class function, and therefore can be used to restrict the pool set to a smaller, yet, impactful set of candidates for annotation.”